# Fear of Cancer Recurrence in Sarcoma Survivors: Results from the SURVSARC Study

**DOI:** 10.3390/cancers14246099

**Published:** 2022-12-11

**Authors:** Ilaria Pellegrini, Cas Drabbe, Dirk J. Grünhagen, Michiel A. J. Van de Sande, Jacco J. de Haan, Kristien B.M.I. Keymeulen, Johannes J. Bonenkamp, Winette T. A. Van der Graaf, Olga Husson

**Affiliations:** 1Department of Medical Oncology, The Netherlands Cancer Institute, 1066 CX Amsterdam, The Netherlands; 2Department of Medical Oncology, IRCCS Istituto Nazionale dei Tumori, 20133 Milan, Italy; 3Department of Medical Oncology, Erasmus MC Cancer Institute, 3015 GD Rotterdam, The Netherlands; 4Department of Surgical Oncology, Erasmus MC Cancer Institute, 3015 GD Rotterdam, The Netherlands; 5Department of Orthopaedic Surgery, Leiden University Medical Center, 2333 ZA Leiden, The Netherlands; 6Department of Medical Oncology, University Medical Center Groningen, 9713 GZ Groningen, The Netherlands; 7Department of Surgical Oncology, Maastricht University Medical Center, 6229 HX Maastricht, The Netherlands; 8Department of Surgical Oncology, Radboud University Medical Center, 6525 GA Nijmegen, The Netherlands; 9Division of Clinical Studies, Institute of Cancer Research, London SM2 5NG, UK

**Keywords:** fear of cancer recurrence, sarcoma, survivors, follow-up, health-related quality of life

## Abstract

**Simple Summary:**

Fear of cancer recurrence is often reported as an unmet concern by cancer patients, and, to our knowledge, it has not been assessed yet in sarcoma survivors. We conducted a population-based cross sectional questionnaire study to assess the prevalence and characteristics of fear of cancer recurrence amongst sarcoma survivors. We demonstrated that severe fear of cancer recurrence is common in sarcoma survivors and that high levels are associated with decreased global health status. Fear of cancer recurrence deserves more attention in optimal sarcoma survivorship care. To guarantee adequate patient care, the collaboration between health care professionals should be encouraged, and structured support programs should be developed to deliver interventions in a correct and time adequate environment.

**Abstract:**

Fear of cancer recurrence (FCR) is often reported as an unmet concern by cancer patients. The aim of our study was to investigate (1) the prevalence of FCR in sarcoma survivors; (2) the factors associated with a higher level of FCR; the relationship between (3) FCR and global health status and (4) FCR and use of follow-up care. Methods: A cross-sectional study was conducted among sarcoma survivors 2 to 10 years after diagnosis. Patients completed the Cancer Worry Scale (CWS), the global health status subscale of the EORTC QLQ-C30 and a custom-made questionnaire on follow-up care. Results: In total, 1047 patients were included (response rate 55%). The prevalence of high FCR was 45%. Factors associated with high FCR were female sex with 1.6 higher odds (95% CI 1.22–2.25; *p* = 0.001); having ≥1 comorbidities and receiving any treatment other than surgery alone with 1.5 (95% CI 1.07–2.05; *p* = 0.017) and 1.4 (95% CI 1.06–1.98; *p* = 0.020) higher odds, respectively. Patients on active follow-up had 1.7 higher odds (95% CI 1.20–2.61; *p* = 0.004) and patients with higher levels of FCR scored lower on the global health status scale (72 vs. 83 *p* ≤ 0.001). Conclusions: Severe FCR is common in sarcoma survivors and high levels are related to a decreased global health status. FCR deserves more attention in sarcoma survivorship, and structured support programs should be developed to deliver interventions in a correct and time adequate environment.

## 1. Introduction

Sarcomas are a heterogeneous group of malignant mesenchymal neoplasms that consist of over 100 histological subtypes defined by distinctive morphological, immunohistochemical and molecular features [1]. They are rare tumours, with an estimated incidence of 4–5 per 100,000 per year in Europe [2], and represent approximately 1% of all cancers [3]. The biological complexity, diversity and rarity of these solid tumours explain why the outcomes in patients with sarcomas are poorer compared with epithelial carcinomas with a 5-year survival rate of 55–60% for soft tissue sarcomas (STS) and 50–55% for bone sarcomas (BS) [4]. 

To provide adequate multimodality care, the clinical management of sarcomas should be carried out in specialized centers or within a network of centers with an appropriate expertise based on a higher number of patients treated annually [5]. Patients treated in centers of expertise more often receive guideline-oriented therapies, have access to clinical studies, and have a prolonged life expectancy [5,6,7]. It is, therefore, not surprising that quality of life and survivorship issues are gaining more attention amongst clinicians and researchers [8,9]. Health-related quality of life (HRQoL) is a multidimensional concept that includes the patient’s perception of the impact of the disease and its treatment on physical/biological, psychological and social functioning [10]. Better knowledge of HRQoL is essential to provide patient-centered care, as cancer patients consider HRQoL to be an important (treatment) outcome [9,11]. Moreover, cured patients and long-term survivors may have long-lasting problems which may be overlooked without a proper HRQoL assessment [12]. 

One of the main problems cancer survivors may encounter is the fear of cancer recurrence (FCR) and this might even be a bigger issue for rare cancer patients, given the many challenges posed by these tumours (i.e., late or incorrect diagnosis; lack of access to appropriate therapies and expertise and difficulties in conducting well-powered clinical studies [13]. FCR is defined as the fear or worry that the disease will return or progress in the same organ or in another part of the body [14]. Some level of symptom awareness and alertness is functional as it keeps the patient watchful of possible symptoms of recurrence. However, a high level of FCR leads to psychological distress, lower HRQoL, and functional impairments [15]. Patients with elevated levels of FCR will more frequently consult health care professionals for reassurance and they will less frequently accept to be discharged from follow-up care [16,17]. This leads to higher costs on public health, busier clinics and longer waiting times. Simard et al. showed that up to 79% of cancer survivors considered FCR to be an unmet concern [18]. There are no widely accepted guidelines for identifying and treating FCR in clinical practice, although several reviews and commentaries provide useful recommendations [19,20,21] and Cancer Australia produced recommendations for the identification and management of FCR in adult cancer survivors [22]. FCR has been studied in different cancer types including breast cancer, prostate cancer, colorectal cancer, gastrointestinal stromal tumours (GIST) [23,24,25,26] and in adolescent and young adult (AYA) cancer patients [27]. To our knowledge, FCR in sarcoma survivors has not been assessed yet.

This study aims to investigate (1) the prevalence of FCR in sarcoma survivors; (2) the factors associated with a higher level of FCR; (3) the relationship between FCR and global health status and (4) the relationship between FCR and the follow up care offered to sarcoma survivors.

## 2. Materials and Methods

### 2.1. Study Design and Participants

The SURVSARC study is a population-based cross-sectional questionnaire study, which included sarcoma survivors (2–10 years after diagnosis) aged ≥18 years who were registered in the Netherlands Cancer Registry (NCR). All participants were diagnosed with sarcoma between 1 January 2008 and 31 December 2016, at one of the six participating sarcoma expertise centers (Radboud University Medical Center (Nijmegen), The Netherlands Cancer Institute (Amsterdam), University Medical Center Groningen, Leiden University Medical Center, Erasmus MC Cancer Institute (Rotterdam), and Maastricht University Medical Center). The exclusion criteria were cognitive impairment, physical condition not well enough as judged by the patient ‘s last treating physician, unverifiable address, or a non-sufficient knowledge of the Dutch language. Patients with desmoid fibromatosis, grade 1 chondrosarcoma, atypical lipomatous tumours, and giant cell tumours were excluded due to the indolent clinical behaviour and subsequent less invasive treatment. In addition, gastrointestinal stromal tumours (GIST) were excluded considering the different treatment strategies compared to other sarcomas. The medical ethical committee of the Radboud University Medical Centre (2017-3944) provided ethical approval, which according to the Dutch law, is valid for all participating centers in case of questionnaire research. The study was registered in the Dutch Trial Registry (NTR-7253).

### 2.2. Recruitment and Data Collection

Eligible patients received a letter from their (former) treating physician, describing the purpose of the study. After providing informed consent, participants could fill the questionnaires either online or on a paper form. Demographic, clinical and treatment characteristics were obtained from the NCR, which contains data from nearly all newly, diagnosed cancer patients in The Netherlands since 1989. Completion of the questionnaires was conducted between October 2018 and June 2019 within the PROFILES (Patient Reported Outcomes Following Initial treatment and Long-term Evaluation of Survivorship) data management system. The study used the NCR as sampling frame which made it possible to perform a non-responder’s analysis, allowing to have some information about the representativeness of our study sample. Further details regarding the data collection methods have been described in previous papers [28,29].

### 2.3. Study Measures

The current study is a pre-planned secondary analysis of the SURVSARC study. A total of 1887 sarcoma survivors were approached to participate in the SURVSARC study, of whom 1099 provided informed consent and completed the questionnaires (response rate 58%). Among the total of survivors approached, 788 patients (42%) did not participate in the study because they did not want to be confronted with the disease, they did not know they had a sarcoma, or they were considered too frail. The information and questionnaires which we considered relevant for our investigation are described in detail below.

#### 2.3.1. Sociodemographic and Clinical Characteristics

Patient, tumour, and treatment characteristics were patient-reported, and any missing data were derived from the NCR, which regularly collects data such as sex, age, date of diagnosis, histological subtype, localization, tumour grade, and stage. In our study, the number and type of comorbidities were pooled into 3 groups (no comorbidities-one comorbidity-two comorbidities or more). Time since diagnosis was calculated by subtracting the date of questionnaire completion from the date of diagnosis. 

#### 2.3.2. Cancer Worry Scale

The Cancer Worry Scale (CWS) is a validated tool to assess worries about developing cancer or developing cancer again and the impact of these concerns on daily functioning. It consists of eight items each rated on a four-point Likert scale ranging from never (1) to almost always (4) [30]. Custers at al. firstly validated the scale in breast cancer patients setting a cut-off score of 14 or higher (sensitivity 77%; specificity 81%) as optimal to detect severe FCR levels [23]. The scale was validated in other adult cancer populations [24,25,26] and in the AYA population [27]. The present study used the same cut-off score to indicate high FCR and the answers to each question were grouped considering points 1 and 2 of the Likert scale together, and 3 and 4 together to improve the power of the statistical analysis.

#### 2.3.3. Global Health Status

Global Health Status was measured through the subscale of the European Organization for Research and Treatment of Cancer (EORTC) Quality of Life Questionnaire-Core 30 (QLQ-C30). The EORTC QLQ-C30 is a set of 30 questions and represents a reliable and valid measure of the quality of life of cancer patients in multicultural clinical research settings [31]. Patients had to value their global health status ranging from “very poor” to “excellent” by answering the following question: “How would you rate your overall quality of life during the past week?”. In our study, raw scores were transformed to a linear scale ranging from 0 to 100, with a higher score representing a better QoL and higher functioning level.

#### 2.3.4. Follow-Up Characteristics

The questionnaire on follow-up (FU) consists of a custom-made set of 11 questions drafted to assess the emotional components (“Do the controls reassure you?”—“To what extent do you feel emotions such as tension or anxiety prior to the checks”) and the practical aspects of patient’s FU (“Do you know how often you should go to the doctor for the follow up?”—“Do you know for how long you should continue your follow up?”). The answers are all categorical and vary from yes and no to multiple choices that have been mostly dichotomized in 3 different groups (Agree-Neutral-Disagree). For each of the questions we assessed the percentage of patients with high and low FCR (calculated as described above).

### 2.4. Statistical Analysis

An anonymous comparative analysis between responders and non-responders was conducted by an NCR employee and non-responder data were not shared with the research team. Descriptive statistics were presented for continuous variables (mean (SD) and median (range)) and categorical variables (*n*° (%)). A comparative analysis between the low FCR and high FCR group was conducted on sociodemographic, tumour, and treatment characteristics and follow-up characteristics. Chi-squared tests were used for categorical variables, whereas independent samples t-tests were used for continuous variables. The variables included in the multivariate analysis were those that were statistically significant in the univariate analyses (*p* < 0.05) between the low FCR and high FCR groups. We did not include the conditional questions in the multivariate analysis that were only completed by survivors still in active follow-up. All statistical analyses were carried out using SPSS Statistics (IBM Corporation, version 26.0, Armonk, NY, USA). *p*-values < 0.05 were considered statistically significant.

## 3. Results

In the SURVSARC study, 1099 of 1887 eligible patients provided informed consent and completed the questionnaires. Of these, 1047 patients completed the CWS questionnaire and were included in this analysis (response rate 55%). The non-responder analyses have been published previously, showing an older age and higher socioeconomic status (SES) for responders. There were no differences with regard to sex, time since diagnosis, and sarcoma subtype (bone sarcoma vs. soft tissue sarcoma) between responders and non-responders [28]. 

### 3.1. Sociodemographic, Tumour and Treatment Characteristics

The majority of all survivors was male (54%) and the median age at diagnosis was 56 years (range 18–90) (Table 1). The median time since diagnosis was 62 months (range 20–135), 75% had STS, and 40% had high-grade disease. Almost half of all patients (42%) received surgery as their only treatment. In the comparative analysis between the high and low FCR group, significantly more female survivors were in the high FCR group compared to the low FCR group (54% vs. 39% *p* ≤ 0.001). The same was seen for AYA patients (18% vs. 15% *p* = 0.006), whereas in the elderly subgroup (>70 years old), more patients were in the low FCR group compared with the high FCR group (17.4% vs. 10.7% *p* = 0.006). The three most frequent comorbidities for both high and low FCR were back pain, high blood pressure, and arthrosis. Patients with ≥2 comorbidities showed more frequent and high levels of FCR (42% vs. 28%; *p* ≤ 0.001). Surgery alone represented the treatment of choice in 36% of survivors with high FCR and in 46.5% of survivors with low FCR (*p* = 0.022). Although not statistically significant, survivors with a high FCR were more often seen with high-grade disease compared with survivors with a low FCR (44% vs. 38%, *p* = 0.064). 

### 3.2. The Cancer Worry Scale

A quarter of all survivors (25%) indicated that they thought about the chances of having cancer (again) often to almost always (Table 2). To the question whether these thoughts affected their mood, 9% of all survivors indicated that it did often or almost always. For 5% of all survivors and 11% of the survivors with high FCR, these thoughts interfered with their ability to do daily activities often or almost always. One fifth (20%) of the survivors worried often or almost always about developing cancer again. Worry is a problem for 10% of all survivors and for 22% of survivors with high FCR. How often survivors worry about family members having cancer was indicated as often or almost always by 11% of participants, and is again higher for survivors with high FCR (21%). The possibility of a new surgery was “quite a bit-very much” of a concern in 13% of survivors, and it was higher in the high FCR group (27%). 

### 3.3. Global Health Status

Patients with high FCR had a statistically significant mean score on the global health status scale lower than the low FCR group (72 vs. 83; *p*-value < 0.001). 

### 3.4. Follow up Characteristics

In total, 77% of all survivors were still undergoing active follow-up (Table 3). Survivors with high FCR were more often undergoing active follow-up compared to survivors with low FCR (84% vs. 73%, *p* < 0.001). Almost all survivors (96%) were satisfied with the frequency of follow-up moments. Survivors with low FCR were slightly more often satisfied with the frequency of follow-up moments in comparison with the high FCR group (98% vs. 94%, *p* = 0.006). Many survivors (88%) agreed to the statement that the follow-up moments reassured them; this was lower (83%) for survivors with high FCR than for survivors with low FCR (92%, *p* < 0.001). Over half of all survivors (56%) experienced emotions such as tension and anxiety prior to the controls, and this was experienced more in the high-FCR-group (67% vs. 48%, *p* < 0.001). Of all survivors, 27% experienced these emotions between 1 week and 1 month before appointment and 40% had it less than a week but more than 1 day before the appointment. Survivors with high FCR experienced these emotions generally during a longer period before the follow-up appointment. Over half (61%) of all survivors agreed with the statement that during the follow-up visits, they pay attention to problems/complaints caused by their sarcoma/treatment, other than whether the sarcoma has grown back or not. 

### 3.5. Factors Associated with High FCR

Female sex was associated with 1.6 higher odds of being in the high FCR group (95% CI 1.22–2.25; *p* = 0.001) (Table 4). Considering the number of comorbidities, the odds of a high level of FCR was 1.5 times higher in the group with 1 or more comorbidities when compared with patients who were otherwise healthy (95% CI 1.07–2.05; *p* = 0.017). Any treatment other than surgery alone gave 1.4 higher odds of being in the high FCR group (95% CI 1.06–1.98; *p* = 0.020). Patients still undergoing active follow-up had 1.7 higher odds of falling into the high-FCR-group (95% CI 1.20–2.61; *p* = 0.004).

## 4. Discussion

This population-based cross-sectional questionnaire study amongst sarcoma survivors aged ≥18 years showed, with a median follow-up of about 5 years after diagnosis, that 45% had severe levels of FCR. Our results also revealed that survivors with high FCR had a lower global health status in comparison with survivors with low FCR. Factors associated with a high FCR were female sex, having more than one comorbidity, having had any treatment other than surgery alone, and being under active follow-up at the time of the questionnaire. 

A growing body of literature is available about cancer survivorship; however, the focus is mainly on common cancers [32,33,34,35,36]. Studies on survivorship in rare cancer patients are scarce and research on FCR in rare cancer survivors is even more limited. A study to collect information on the nature of FCR in sarcoma patients, who may be on or off treatment, is currently ongoing in the Sarcoma UK group, aiming to develop an intervention to manage FCR [37]. To our knowledge, the current study is the first one examining FCR in sarcoma survivors. We reported a prevalence of severe levels of FCR of 45%, which was higher compared with more common cancers that used the CWS for the evaluation. In fact, in breast and colorectal cancers the prevalence of high FCR was 31% and 38%, respectively, using the same cut-off score to differentiate between high and low FCR (low: ≤13, high: ≥14), whereas in prostate cancer, the prevalence was 36% using a slightly lower cut off score (low: ≤12, high: ≥13) [23,25,26]. A potential explanation to this variation across cancer types could be that we included patients with various histology, stage of disease and consequently prognoses, whereas other studies included only patients treated with curative intent. A study on FCR in localized or metastatic GIST patients reported a prevalence of 52% which is closer to our findings [24]. The higher prevalence seen in our study is reasonable considering the many unknowns on diagnosis and treatment in patients with sarcoma.

This study showed that HRQoL, assessed through the EORTC QLQ C30 subscale on global health/quality of life, was worse for patients with higher FCR. This result is in line with existing literature where in different cancer types, higher FCR was related to both poorer physical and mental HRQoL [14,38]. It remains unclear if a poorer HRQoL facilitates the development of FCR or if a higher FCR leads to a worse HRQoL.

Female gender, higher number of comorbidities and multimodal treatment were all associated with approximately 1.5 higher risk of having dysfunctional levels of FCR. These same factors were analyzed in a systematic review on FCR in adult cancer survivors reporting similar results concerning gender and contradictory evidence in regard of comorbidities and treatment [18]. Past studies showed that high FCR is more prevalent among AYAs than cancer patients of mixed ages [23,24,25,26,27]. The reasons for the higher prevalence in younger people with cancer are not well studied, however it could be hypothesized that being diagnosed with cancer at a time when most AYAs are trying to plan their career and start a family could make future seem uncertain and increase FCR. We looked at differences in age groups and found a higher percentage of AYAs with high FCR in the univariate analysis (18.4%); however, this result did not hold up in the multivariate analyses, preventing us from further consideration on the age-related impact on FCR. However, in a previous SURVSARC analysis, it was seen that AYAs more often had multimodal treatment and chemotherapy when compared with other age groups [39]. The more aggressive approach to the disease could be related with AYAs’ higher FCR. Interestingly, we did not find differences between grade and the histology of the tumour which are two objective prognostic indicators. A possible explanation could be that treatment is more readily interpreted by patients as signifying serious disease than are tumour characteristics. Furthermore, this result underlines two issues, namely, (1) the lack of patient’s awareness on factors considered when assessing the likelihood of relapse and the routine follow-up policy, and (2) the need for health care professionals to provide complete and understandable information on treatment and prognosis to allow the patient to deal and cope with the cancer diagnosis. 

The idea of focusing on the correlation between FCR and follow-up care derived from the common knowledge that medical consultations can make patients very anxious and uncertain and this psychological distress could relate with FCR [24]. In our analysis, patients still undergoing active follow up showed higher FCR, as expected. Reasonably, these patients have a shorter time interval since diagnosis, and they are aware that the chances of a recurrence decrease with time. However, medical consultation represents a constant reminder of the disease and of the risk of relapse. As a result, the challenge for both oncology professionals and patients is to find a balance between the medical need for follow-up and the psychological issues associated with these visits. 

One of the limitations of our study concerns the use of the CWS to differentiate between high and low FCR. The CWS was not validated in rare cancer survivors, and this may overestimate or underestimate the proportion of patients with sarcomas who experience FCR. Given the cross-sectional character of the study, it is unknown how many patients had a recurrence at the time of study participation, and it is impossible to draw any conclusions on the development of FCR over time. Moreover, there could be a possible selection bias since it is unknown whether non-participating survivors did not participate due to either poor health or an absence of symptoms.

Our results showed that FCR represents an issue in a substantial part of sarcoma survivors. During the follow-up visits, clinicians mostly focus on symptoms and signs of recurrence, considering their absence reassuring enough for the patients. However, other components play a role in the global well-being of a sarcoma survivor, with FCR being one aspect that must be considered when dealing with survivorship. Doctors and nurses should normalize feelings of FCR, and patients should be provided with skills to address FCR early, which may have value in preventing the development of severe FCR. Health care providers (HCPs) should be taught to recognize FCR, explore it and manage it based upon its severity. Moreover, the collaborations with psychosocial care teams either in the hospital or trained in the community is pivotal to support the patients. A recent review by Liu et al. underlined the need for the development of a stepped-care model for managing FCR. For example, patients with mild to moderate FCR could be adequately managed using brief interventions delivered by nurses, general practitioners, and oncologists, whereas more severe cases will need a professional psychological intervention [19]. Common psychotherapeutic interventions for FCR are Cognitive Behavioral Therapy (CBT), which aims to change unhelpful thoughts and feelings [40], and Acceptance and Commitment Therapy (ACT) which emphasizes acceptance and promotes psychological flexibility in managing life’s stressors. This last approach represents a feasible and promising treatment in breast cancer survivors with clinical FCR, reducing maladaptive coping while facilitating adaptive management of FCR [41]. With regards to rare cancers, support from the HCPs is even more essential considering the limited information available and the lack of social comparators. Considering doctor’s pressured follow-up clinic schedule, an effort should be made to build structured programs within the oncological services to deliver adequate support to patients experiencing FCR. Organizational incentives should be given to allow the arrangement of this services across tumour groups to keep care as efficient as possible.

Future research of a qualitative nature should explore what exactly patients fear most and why, to give HCPs more information and knowledge to better approach this issue. The questionnaires currently available do not address the nature of the fear and cannot determine what aspects of recurrence is feared. Research on patients’ views supports the patient-centered model of care, where the patient is an active and autonomous agent who participates in the decision-making processes [42,43]. Moreover, longitudinal studies on FCR could give information about the time needed to develop FCR and its changes over time.

## 5. Conclusions

In conclusion, high FCR is a substantial issue (45%) in the sarcoma survivor population, even years after diagnosis, and high levels of FCR are strongly associated with decreased global health status. These results confirm that FCR deserves more attention in optimal sarcoma survivorship care and, moving towards a patient’s tailored care, factors that trigger FCR and patient’s point of views should be considered when designing follow up programs for sarcoma survivors. A collaboration between HCPs is pivotal to guarantee adequate patient’s care, and structured support programs should be encouraged allowing each specialist to deliver intervention in a correct and time-adequate environment.

## Figures and Tables

**Table 1 cancers-14-06099-t001:** Sociodemographic, tumour, and treatment characteristics.

	Total *n* = 1047	High FCR *n* = 467	Low FCR *n* = 580	*p*-Value
	*n*° (%)	*n*° (%)	*n*° (%)	
Sex				
Male	566 (54.1)	213 (45.6)	353 (60.9)	<0.001
Female	481 (45.9)	254 (54.4)	227 (39.1)	
Age at diagnosis in years				0.023
Mean (± SD)	54.5 (15.2)	53.3 (14.5)	55.5 (15.7)	
Median (range)	56 (19–90)	55 (19–90)	57 (18–87)	
AYA (18–39 years) Older adults (40–69 years) Elderly (>70 years)	174 (16.6) 722 (69.0) 151 (14.4)	86 (18.4) 331 (70.9) 50 (10.7)	88 (15.2) 391 (67.4) 101 (17.4)	0.006
Age at questionnaire in years				
Mean (±SD)	60.2 (15.0)	58.8 (14.5)	61.2 (15.3)	0.008
Median (range)	62 (21–94)	61 (22–94)	63 (21–92)	
Time since diagnosis in months				
Mean (±SD) Median (range)	67.7 (30.5) 62(20–135)	65.9 (30.6) 59 (21–135)	69.1 (30.4) 65 (20–135)	0.086
Comorbidities				
0	351 (33.5)	136 (29.1)	215 (37.1)	<0.001
1	337 (32.2)	136 (29.1)	201 (34.7)	
≥2	359 (34.3)	195 (41.8)	164 (28.3)	
Histology				STS versus BS 0.933 Histological subtypes 0.063
STS	786 (75.1)	350 (74.9)	436 (75.2)	
DFSP	68 (6.5)	17 (3.6)	51 (8.8)	
Leiomyosarcoma	106 (10.1)	50 (10.7)	56 (9.7)	
Liposarcoma	167 (16.0)	70 (15.0)	97 (16.7)	
Myxofibrosarcoma	131 (12.5)	53 (11.3)	78 (13.4)	
MPNST	31 (3.0)	19 (4.1)	12 (2.1)	
Rhabdomyosarcoma	14 (1.3)	9 (1.9)	5 (0.9)	
Synovial sarcoma	34 (3.2)	14 (3.0)	20 (3.4)	
Vascular sarcoma	42 (4.0)	21 (4.5)	21 (3.6)	
Other STS	193 (18.4)	97 (20.8)	96 (16.6)	
BS	261 (24.9)	117 (25.1)	144 (24.8)	
Chondrosarcoma	124 (11.8)	56 (12.0)	68 (11.7)	
Chordoma	28 (2.7)	12 (2.6)	16 (2.8)	
Ewing sarcoma	28 (2.7)	12 (2.6)	16 (2.8)	
Osteosarcoma	68 (6.5)	32 (6.9)	36 (6.2)	
Other BS	13 (1.2)	5 (1.1)	8 (1.4)	
Grade				
Low grade	580 (59.6)	244 (56.4)	336 (62.2)	0.064
High grade	393 (40.4)	189 (43.6)	204 (37.8)	
Missing	74	34	40	
Clinical staging				
Stage I	456 (48.3)	178 (42.8)	278 (52.6)	0.016
Stage II	304 (32.2)	154 (37.0)	150 (28.4)	
Stage III	128 (13.5)	59 (14.2)	69 (13.0)	
Stage IV	57 (6.0)	25 (6.0)	32 (6.0)	
Missing	102	51	51	
Localization				
Head and Neck	64 (6.1)	30 (6.4)	34 (5.9)	0.001
Thoracic	79 (7.5)	40 (8.6)	39 (6.7)	
Abdominal	127 (12.1)	65 (13.9)	62 (10.7)	
Breast	24 (2.3)	17 (3.6)	7 (1.2)	
Skin	103 (9.8)	28 (6.0)	75 (12.9)	
Pelvis	79 (7.5)	38 (8.1)	41 (7.1)	
Upper extremities	110 (10.5)	46 (9.9)	64 (11.0)	
Lower extremities	392 (37.4)	166 (35.5)	226 (39.0)	
Other	69 (6.6)	37 (7.9)	32 (5.5)	
Treatment				
Surgery only	437 (41.8)	168 (36.0)	269 (46.5)	0.022 ^a^
RT only	15 (1.4)	9 (1.9)	6 (1.0)	
CT only	8 (0.8)	4 (0.9)	4 (0.7)	
Surgery and RT	411 (39.3)	198 (42.4)	213 (36.8)	
Surgery and CT	78 (7.5)	35 (7.5)	43 (7.4)	
RT and CT	11 (1.1)	6 (1.3)	5 (0.9)	
Surgery and RT and CT	86 (8.2)	47 (10.1)	39 (6.7)	
Missing	1	0	1	

^a^ Fisher’s exact. AYA = Adolescent and Young Adult, STS = soft-tissue sarcoma, BS = bone sarcoma, DFSP = dermatofibrosarcoma protuberans, MPNST = Malignant Peripheral Nerve Sheath Tumor.

**Table 2 cancers-14-06099-t002:** The cancer worry scale.

	Total *n* = 1047	High FCR *n* = 467	Low FCR *n* = 580	*p*-Value
	*n*° (%)	*n*° (%)	*n*° (%)	
1. How often have you thought about your chances of getting cancer (again)?				
Almost never—Sometimes	788 (75.3)	225 (48.2)	563 (97.1)	<0.001
Often—Almost always	259 (24.7)	242 (51.8)	17 (2.9)	
2. Have these thoughts affected your mood?				
Almost never—Sometimes	955 (91.2)	377 (80.7)	578 (99.7)	<0.001
Often—Almost always	92 (8.8)	90 (19.3)	2 (0.3)	
3. Have these thoughts interfered with your ability to do daily activities?				
Almost never—Sometimes	994 (94.9)	416 (89.1)	578 (99.7)	<0.001
Often—Almost always	53 (5.1)	51 (10.9)	2 (0.3)	
4. How concerned are you about the possibilities of getting cancer (again) one day?				
Not at all—a little	791 (75.5)	220 (47.1)	571 (98.4)	<0.001
Quite a bit—Very much	256 (24.5)	247 (52.9)	9 (1.6)	
5. How often do you worry about developing cancer (again)?				
Almost never—Sometimes	842 (80.4)	263 (56.3)	579 (99.8)	<0.001
Often—Almost always	205 (19.6)	204 (43.7)	1 (0.2)	
6. How much of a problem is this worry?				
Not at all—a little	941 (89.9)	364 (77.9)	577 (99.5)	<0.001
Quite a bit—very much	106 (10.1)	103 (22.1)	3 (0.5)	
7. How often do you worry about the chance of family members developing cancer?				
Almost never—Sometimes	931 (88.9)	367 (78.6)	564 (97.2)	<0.001
Often—Almost always	116 (11.1)	100 (21.4)	16 (2.8)	
8. How concerned are you about the possibility that you will ever need surgery (again)?				
Not at all—a little	912 (87.1)	342 (73.2)	570 (98.3)	<0.001
Quite a bit—Very much	135 (12.9)	125 (26.8)	10 (1.7)	

**Table 3 cancers-14-06099-t003:** Follow-up characteristics.

All Survivors	Total *n* = 1047	High FCR *n* = 467	Low FCR *n* = 580	*p*-Value
	*n*° (%)	*n*° (%)	*n*° (%)	
1. Are you still undergoing active follow-up?				
Yes	656 (77.1)	288 (83.7)	368 (72.6)	<0.001
No	195 (22.9)	56 (16.3)	139 (27.4)	
Missing	196	123	73	
Survivors still undergoing active follow-up	Total *n* = 656	High FCR *n* = 288	Low FCR *n* = 368	*p*-value

	*n*° (%)	*n*° (%)	*n*° (%)	
2. Is there an agreement about how often you should now return to your specialist?				
Yes	629 (96.3)	276 (96.2)	353 (96.4)	0.850
No	24 (3.7)	11 (3.8)	13 (3.6)	
Missing	3	1	2	
3. Is there an agreement about how long you should continue active follow-up (calculated from the moment that you finished your (first) treatment)?				
Yes, less than 5 years	47 (7.7)	14 (5.3)	33 (9.4)	0.162
Yes, more than 5 years	393 (64.0)	174 (65.9)	219 (62.6)	
No	174 (28.3)	76 (28.8)	98 (28.0)	
Missing	42	24	18	
4. Are you satisfied with the frequency of the follow-up appointments?				
Yes	628 (96.0)	269 (93.7)	359 (97.8)	0.015 ^a^
No, I want follow-up appointments more often	17 (2.6)	14 (4.9)	3 (0.8)	
No, I want follow-up appointments less often	7 (1.1)	3 (1.0)	4 (1.1)	
No, I don’t want any follow-up appointments	2 (0.3)	1 (0.3)	1 (0.3)	
Missing	2	1	1	
5. Do the follow-up appointments reassure you?				
Yes, very much	578 (88.5)	239 (83.6)	339 (92.4)	0.001 ^a^
Neutral	64 (9.8)	42 (14.7)	22 (6.0)	
Not at all	11 (1.7)	5 (1.7)	6 (1.6)	
Missing	3	2	1	
6. (To what extent) do you feel emotions such as tension or anxiety prior to the follow-up appointments?				
A lot	366 (56.0)	191 (66.8)	175 (47.7)	<0.001
Neutral	141 (21.6)	81 (28.3)	60 (16.3)	
None	146 (22.4)	14 (4.9)	132 (36.0)	
Missing	3	2	1	
7. How long before the follow-up appointments do you feel those emotions?				
From the previous follow-up appointment	9 (1.8)	9 (3.3)	0 (0.0)	<0.001 ^a^
More than a month	14 (2.8)	9 (3.3)	5 (2.1)	
Between 1 week and 1 month	136 (26.9)	103 (37.9)	33 (14.1)	
Less than 1 week but more than 1 day	204 (40.3)	101 (37.1)	103 (44.0)	
At most 1 day before	143 (28.3)	50 (18.4)	93 (39.7)	
Missing	150	16	134	
8. If you get a good result during a follow-up appointment, (to what extent) do you feel emotions such as tension or anxiety after the control?				
A lot	199 (30.7)	108 (37.9)	91 (25.1)	<0.001
Neutral	88 (13.6)	57 (20.0)	31 (8.5)	
None	361 (55.7)	120 (42.1)	241 (66.4)	
Missing	8	3	5	
9. How long after the follow-up appointment do you feel those emotions?				
From the previous follow-up appointment	14 (4.9)	9 (5.5)	5 (4.1)	<0.001 ^a^
More than a month	6 (2.1)	6 (3.6)	0 (0.0)	
Between 1 week and 1 month	30 (10.5)	23 (13.9)	7 (5.8)	
Less than 1 week but more than 1 day	105 (36.7)	75 (45.5)	30 (24.8)	
At most 1 day before	131 (45.8)	52 (31.5)	79 (65.3)	
Missing	370	123	247	
10. During the follow-up appointment, attention is paid to problems/complaints caused by the sarcoma/treatment, other than whether it has recurred?				
Yes	400 (61.5)	169 (59.1)	231 (63.5)	0.215
Sometimes	132 (20.3)	67 (23.4)	65 (17.9)	
No	118 (18.2)	50 (17.5)	68 (18.7)	
Missing	6	2	4	
11. Have you received different types of care for your sarcoma other than the medical one (i.e., psychologist, general practitioner, physiotherapist)?				
Yes	393 (37.9)	198 (42.9)	195 (33.9)	0.003
No	644 (62.1)	264 (57.1)	380 (66.1)	
Missing	10	5	5	

^a^ Fisher’s exact.

**Table 4 cancers-14-06099-t004:** Multivariate logistic regression analysis for factors associated with high FCR.

Risk Factors	Odds Ratio	95% CI	*p*-Value
Female sex	1.664	1.229–2.255	0.001
AYA	1.332	0.896–1.981	0.156
≥1 comorbidities	1.483	1.072–2.051	0.017
Stage II or higher	1.207	0.887–1.642	0.232
Any treatment other than surgery alone ^a^	1.453	1.061–1.988	0.020
Ongoing active follow up	1.776	1.207–2.612	0.004
Supportive care	1.242	0.906–1.702	0.179

Multivariate logistic regression analysis amongst sarcoma survivors for the odds of (1) having high FCR vs. (0) low FCR. ^a^ The odds ratio for having undergone multimodal treatment (RT or CT, combined or alone, and with or without surgery) vs. surgery only. AYA = Adolescent and Young Adult.

## Data Availability

The data presented in this study are available on request from the PROFILES registry. The raw data are not publicly available due to privacy.

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
