# Peer review of "Fear of Cancer Recurrence in Sarcoma Survivors: Results from the SURVSARC Study"

_cancers, 2022, doi:10.3390/cancers14246099_

Round 1
Reviewer 1 Report
Dear authors, thank you for having deepened this argument in this population, since this is an important concern for patients. Your research can give some useful suggestions to inform clinical practise.
MATHERIALS AND METHODS:
-Line 98: I would suggest you consider if use a more current terminology to define the study design, to clarify it better to the reader. I suggest you insert the definition of “cross-sectional study” earlier in the paper, since now you have inserted it in the discussion.
-Line 109: please explain the acronymous GIST and be careful to do not refer to patients using the name of their pathology.
-Line 127: it would be interesting to the reader to know the methods of approach to participants, and the specific reasons why patients were not included in the study. It would be nice to represent the entire process with a diagram.
Did you consider any strategy to address potential sources of bias? If yes, please add this aspect.
-Line 145: I am not sure to have exactly understood this paragraph. At line 143 you assert that in the first validation of the scale authors fixed a cut-off score of 13 versus 14, while in your study you used “a cut-off score of 14 or more to 145 indicate high FCR” why did you made it? Can you please add some explanation of this choice?
-Line 152: to be consistent with the previous paragraph can you please add some information about the number of items that compose the scale and on the scoring methods?
-Line 163: can you please add some information about the psychometric characteristics of the scale, as you made for the other variables?
RESULTS:
PARAGRAPH 3.2: In this paragraph you make a comment on seven of the eight questions of the scale, it would be more complete to add a phrase about what merged from the last question.
-Line 219: you missed to add the full stop at the end of the phrase.
DISCUSSION:
-Line 253: here you refer only to adults, but previously you mentioned the AYA within your study population.
Author Response
-Line 98: as you suggested, I inserted "cross-sectional study" in the methods
-Line 109: the acronymous has been already explained in line 89 (Introduction), however to clarify our methods, we decided to add it to the method section too.
-Line 127: The details are described in a previous paper (28 in the bibliography), however we decided to add some more information to our manuscript to improve the readability.
We conducted a population-based study by using the NCR as sampling frame. This made it possible to conduct a non-responders analysis, so at least we have some information about the representativeness of our study sample
-Line 145: We reworded the text to make it clear. We used the same cut-off of the validation paper by Custers et al.
-Line 152: We have added this information to the measure section.
-Line 163: The follow-up questionnaire consists of single items. No psychometric scale properties can be presented.
Paragraph 3.2: I added a last sentence about the last question, as you suggested
-Line 219: Adapted accordingly
-Line 253: We adapted the text to make it clear
Reviewer 2 Report
Pellegrini et al in the article titled `Fear of Cancer Recurrence in sarcoma survivors: results from SURVSARC study` include very detailed statistical description of different age, sex and different FCR scale sarcoma survivors FCR.
The article is well written and the tables are of very good quality.
It also stresses the importance of providing the sarcoma survivors with the additional help with anxiety in form of CBT for example. The authors also stress how much FCR could burden the survivors as well as the whole health system of the country.
Author Response
We improved the English language and checked for any spelling mistakes.
We are very pleased for your positive comments.